# Identification of Pol IV and RDR2-dependent precursors of 24 nt siRNAs guiding de novo DNA methylation in Arabidopsis

Todd Blevins[1,2,3†‡], Ram Podicheti[4,5†], Vibhor Mishra[2,3], Michelle Marasco[2,3], Jing Wang[2,3], Doug Rusch[4], Haixu Tang[5], Craig S Pikaard[1,2,3*]

[1]Howard Hughes Medical Institute, Indiana University, Bloomington, United States; [2]Department of Biology, Indiana University, Bloomington, United States; [3]Department of Molecular and Cellular Biochemistry, Indiana University, Bloomington, United States; [4]Center for Genomics and Bioinformatics, Indiana University, Bloomington, United States; [5]School of Informatics and Computing, Indiana University, Bloomington, United States

*For correspondence: cpikaard@indiana.edu

[†]These authors contributed equally to this work

Present address: [‡]Institut de Biologie Moléculaire des Plantes du Centre national de la recherche scientifique, Université de Strasbourg, Strasbourg, France

**Competing interests:** The authors declare that no competing interests exist.

**Abstract** In *Arabidopsis thaliana*, abundant 24 nucleotide small interfering RNAs (24 nt siRNA) guide the cytosine methylation and silencing of transposons and a subset of genes. 24 nt siRNA biogenesis requires nuclear RNA polymerase IV (Pol IV), RNA-dependent RNA polymerase 2 (RDR2) and DICER-like 3 (DCL3). However, siRNA precursors are mostly undefined. We identified Pol IV and RDR2-dependent RNAs (P4R2 RNAs) that accumulate in *dcl3* mutants and are diced into 24 nt RNAs by DCL3 in vitro. P4R2 RNAs are mostly 26-45 nt and initiate with a purine adjacent to a pyrimidine, characteristics shared by Pol IV transcripts generated in vitro. RDR2 terminal transferase activity, also demonstrated in vitro, may account for occasional non-templated nucleotides at P4R2 RNA 3' termini. The 24 nt siRNAs primarily correspond to the 5' or 3' ends of P4R2 RNAs, suggesting a model whereby siRNAs are generated from either end of P4R2 duplexes by single dicing events.

## Introduction

In plants, transposable elements, transgenes, repetitive sequences and endogenous genes can be transcriptionally silenced by RNA-directed DNA methylation (*Matzke and Mosher, 2014*; *Pikaard et al., 2013*; *Wierzbicki, 2012*; *Zhang and Zhu, 2011*). In this process, 24 nucleotide small interfering RNAs (24 nt siRNAs) bound to an Argonaute family protein, primarily Argonaute 4 (AGO4; [*Zilberman et al., 2003*]), guide the cytosine methylation and histone modification of corresponding DNA sequences, leading to chromatin states that are refractive to transcription by RNA polymerases I, II, or III (*Figure 1*). Approximately 4000–8000 loci in *Arabidopsis thaliana* give rise to clusters of 24 nt siRNAs, collectively accounting for more than 90% of the total small RNA pool (*Law et al., 2013*; *Mosher et al., 2008*; *Zhang et al., 2007*).

Three RNA polymerases are critical for production of non-coding RNAs guiding RNA-directed DNA methylation. These enzymes are nuclear RNA polymerase IV (Pol IV; *Herr et al., 2005*; *Onodera et al., 2005*), nuclear RNA polymerase V (Pol V; *Kanno et al., 2005*; *Pontier et al., 2005*) and RNA-dependent RNA polymerase 2 (RDR2) (*Xie et al., 2004*). Pol IV and Pol V are 12-subunit enzymes (*Ream et al., 2009*) that evolved as specialized forms of DNA-dependent RNA Pol II (*Haag et al., 2014*; *Haag and Pikaard, 2011*; *He et al., 2009*; *Huang et al., 2009*; *Lahmy et al.,*

**eLife digest** Genes contain instructions for processes in cells and therefore their activities must be carefully controlled. The addition of small chemical tags called methyl groups to DNA is one of the many ways by which cells can influence gene activity. These methyl groups can silence genes by altering the DNA so that is more tightly packed within the nucleus of the cell. Virus genes and mobile sections of DNA called transposable elements (sometimes known as jumping genes) are also silenced by DNA methylation to keep them from doing harm.

In plants, methyl groups can be attached to DNA by proteins that are guided to the DNA by molecules called short interfering ribonucleic acids (or siRNAs for short). Each siRNA is made of a chain of 24 building blocks called nucleotides and is able to bind to matching RNA molecules that are attached to the target DNA. The siRNAs are made from longer RNA molecules in a process that involves trimming by an enzyme called DCL3. However, it is not clear how long these "precursor" molecules are before DCL3 cuts them down to size.

Here, Blevins, Podicheti et al. studied how siRNAs are made in a plant called *Arabidopsis thaliana*. The experiments show that RNAs containing around 26-45 nucleotides accumulate in cells that lack DCL3 and these cells are unable to make 24 nucleotide long siRNAs. Furthermore, the purified DCL3 enzyme can cut these precursor RNAs to make the siRNAs. Because the precursors are relatively short, the experiments suggest that DCL3 only cuts each precursor RNA once when making siRNAs.

Blevins, Podicheti et al. also show that the siRNA precursors are made by a partnership of two RNA synthesizing enzymes. Therefore, a challenge for the future will be to understand exactly how they work together.

*2009*; *Ream et al., 2009*). RDR2 is one of six single-subunit RNA-dependent RNA polymerases in Arabidopsis (*Wassenegger and Krczal, 2006*; *Xie et al., 2004*). Pol IV and RDR2 are each essential for the biogenesis of 24 nt siRNAs and they physically associate in Arabidopsis and maize (*Haag et al., 2012*, *2014*; *Law et al., 2011*), suggesting that their activities are coupled for the production of double-stranded siRNA precursors from initial DNA templates. Pol V is not required for siRNA biogenesis at most loci (*Mosher et al., 2008*) but generates non-coding RNAs to which siRNA-AGO4 complexes bind (*Wierzbicki et al., 2008*; *2009*). The C-terminal domain of the Pol V largest subunit also interacts with AGO4 (*El-Shami et al., 2007*). Together, these RNA and protein interactions bring AGO4 to the vicinity of the DNA transcribed by Pol V, allowing recruitment of DNA methylation and chromatin-modifying activities that bring about transcriptional gene silencing (*Figure 1*).

Genetic evidence suggests that coordinated Pol IV and RDR2 transcription yields double-stranded RNAs (dsRNAs) that are then cleaved by DICER-like 3 (DCL3) to produce 24 nt siRNA duplexes, one strand of which is incorporated into AGO4 (*Xie et al., 2004*; *Zilberman et al., 2003*) or a closely related Argonaute protein (*Mallory and Vaucheret, 2010*; *Zheng et al., 2007*). Because Pol IV localization is unaffected in *rdr2* null mutants, but RDR2 is mis-localized in Pol IV null mutants (*Pontes et al., 2006*), Pol IV has been thought to act first in the pathway, generating single-stranded RNAs that then serve as templates for second strand synthesis by RDR2 (*Pikaard, 2006*). Current models have also presumed that dsRNAs made by Pol IV and RDR2 are long, perfectly paired duplexes that can be diced into multiple siRNAs, as depicted in *Figure 1*.

Pol IV and RDR2 will synthesize short transcripts from oligonucleotide templates in vitro (*Haag et al., 2012*), but the sizes of transcripts they are capable of generating is unclear, in vitro or in vivo. Recently, Pol IV and RDR2-dependent transcripts that accumulate in vivo in *dcl2 dcl3 dcl4* (*dcl2/3/4*) mutants were identified (*Li et al., 2015*). In that study, RNAs were fragmented prior to sequencing and short sequence reads that overlapped were then assembled, computationally, into longer contiguous sequences. These analyses led to the conclusion that Pol IV and RDR2 generate transcripts that can be many hundreds of nucleotides in length, and thus large enough to encode multiple siRNAs, as in current models. Here, we present RNA blot hybridization, deep sequencing, in vitro transcription and in vitro DCL3 dicing results that yield a different interpretation, showing that Pol IV and RDR2 primarily generate dsRNAs shorter than 45 nucleotides. Based on sequence

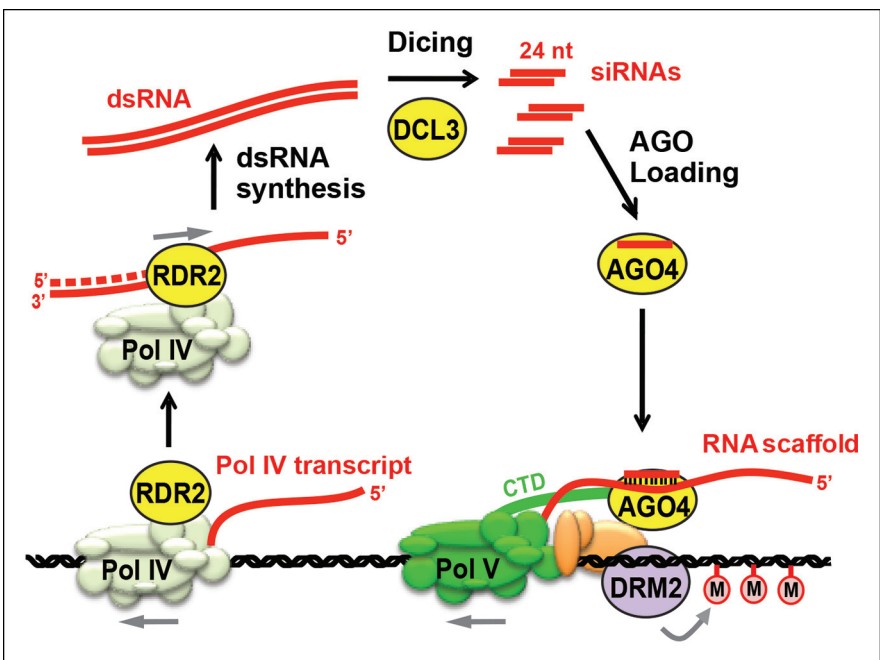

**Figure 1.** Biogenesis of 24 nt siRNAs and their role in RNA-directed DNA methylation. A simplified cartoon of the RNA-directed DNA methylation pathway. Polymerase (Pol) IV and RNA-dependent RNA polymerase (RDR2) physically associate and are required for the synthesis of double-stranded RNAs (dsRNA) that are diced by DICER-like 3 (DCL3) into 24 nt siRNA duplexes. Upon loading into Argonaute 4 (AGO4), the siRNA-AGO4 complex finds its target sites by binding to Pol V transcripts and by interacting with the C-terminal domain (CTD) of the Pol V largest subunit. The cytosine methyltransferase DRM2 is ultimately recruited to Pol V-transcribed loci, resulting in de novo cytosine methylation in all sequence contexts (CG, CHG and CHH; where H represents a nucleotide other than G).

---

alignments and sequence motifs shared by Pol IV/RDR2-dependent RNAs (P4R2 RNAs) and siRNAs, we propose that siRNAs are typically generated by a single internal DCL3 cleavage event whose position is measured from either end of a short P4R2 RNA.

## Results

### siRNA precursors accumulate in mutants defective for DCL3

Previously published RNA blot analyses have shown that in *dcl3* null mutants, loss of 24 and 23 nt siRNAs is accompanied by accumulation of RNAs longer than 25 nt but typically shorter than 50 nt (*Blevins et al., 2009*; *Daxinger et al., 2009*; *Henderson et al., 2006*; *Pontes et al., 2006*). A prime example is the intergenic region that separates tandemly arranged 5S ribosomal RNA (5S rRNA) genes, where RNAs of ~26–45 nt accumulate in *dcl3* null mutants (*Figure 2A*, lanes 3,5,7,8). These longer RNAs accumulate to their highest levels in *dcl2/3/4* triple mutants, in which multiple siRNA processing pathways are impaired (*Figure 2A*, lane 8; see also [*Henderson et al., 2006*]). The longer RNAs that accumulate in *dcl2/3/4* triple mutants fail to accumulate in the absence of Pol IV (*nrpd1* subunit mutant) or RDR2 (lanes 9,10). This indicates that Pol IV and RDR2 are each required for the biogenesis of the longer RNAs, paralleling their requirement for 24 and 23 nt siRNA biogenesis (lanes 13,14). The longer RNAs (>25 nt) are also reduced in abundance if Pol V or AGO4 are mutated in the *dcl2/3/4* triple mutant background (lanes 11,12), analogous to the degree to which siRNA levels are reduced, but not eliminated, in *ago4* or *pol V* single mutants (lanes 15,16). These observations are consistent with Pol V and AGO4 being important, but not essential, for biogenesis of the majority of 23/24 nt siRNAs (*Mosher et al., 2008*; *Pontier et al., 2005*), most likely due to their roles in establishing or maintaining DNA and histone methylation marks that recruit Pol IV to its sites of action (*Blevins et al., 2014*; *Law et al., 2013*; *Zhang et al., 2013*).

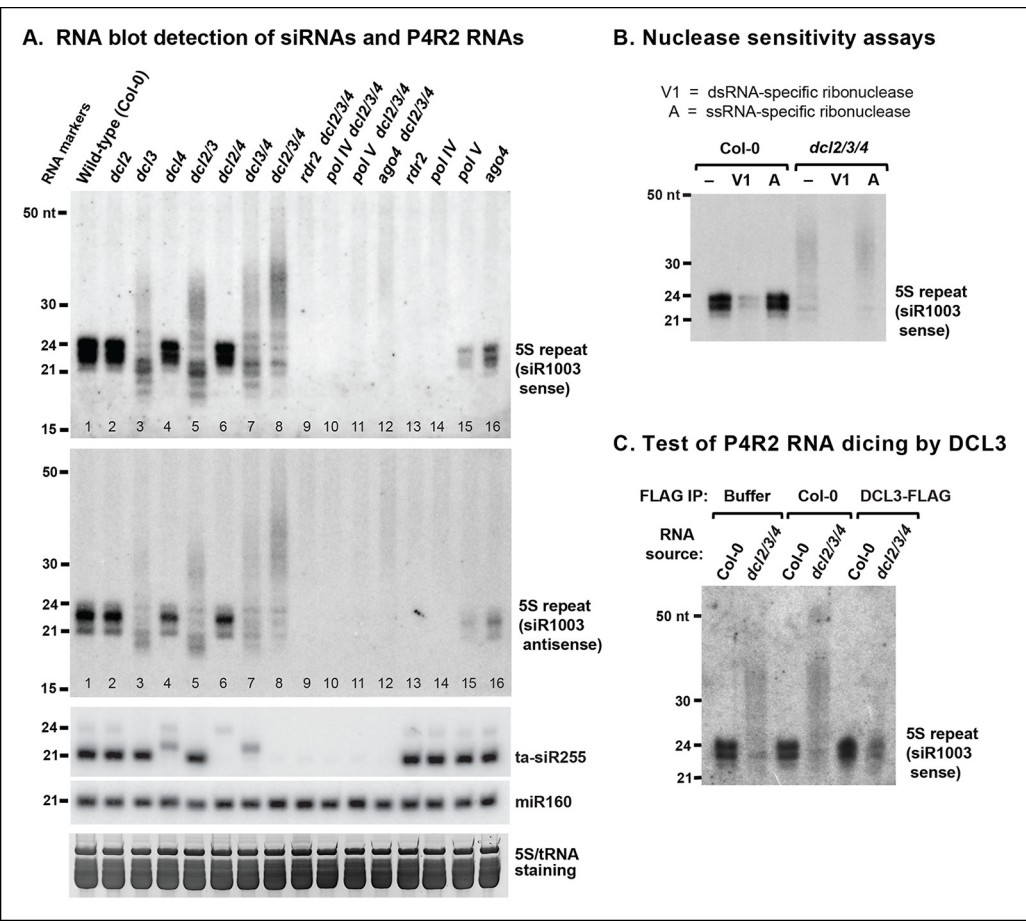

**Figure 2.** RNA blot analyses of 24 nt siRNAs and their precursors. (**A**) The small RNA blot was successively hybridized to probes representing either strand of the siR1003 duplex, a small interfering RNA (siRNA) that is derived from intergenic regions separating 5S ribosomal RNA (rRNA) gene repeats (top two images), as well as to a *trans*-acting siRNA (ta-siR255) and a microRNA (miR160) probe. An image of the stained gel under fluorescent illumination (in the region that includes 5S rRNA and transfer RNAs [tRNAs]) is shown at the bottom as a loading control. (**B**) RNA blot of small RNAs isolated from wild-type (ecotype Col-0) or *dcl2 dcl3 dcl4* triple mutant (*dcl2/3/4*) plants, with or without prior treatment with ribonuclease V1 or ribonuclease A. The blot was hybridized to a probe designed to detect siR1003 'sense', which arises from 5S rRNA gene intergenic spacers. (**C**) Dicing of precursor RNAs by DICER-like 3 (DCL3) in vitro. RNA isolated from wild-type (ecotype Col-0) or from *dcl2/3/4* triple mutant plants was incubated with anti-FLAG resin that had been incubated with protein extraction buffer, a cell-free extract of wild-type (Col-0) plants, or a cell-free extract of transgenic plants expressing FLAG-tagged DCL3. RNAs were then purified, subjected to blotting and hybridized to the siR1003 'sense' probe.

Pol IV and RDR2-dependent >25 nt RNAs (hereafter named P4R2 RNAs) hybridize to 5S gene intergenic region probes corresponding to either DNA strand, as do homologous siRNAs (*Figure 2A*; top two panels). Moreover, the P4R2 RNAs are more sensitive to ribonuclease V1, which preferentially degrades dsRNAs, than to ribonuclease A, which preferentially digests single-stranded RNAs (*Figure 2B*). Interestingly, 23 and 24 nt siRNA signals are reduced upon ribonuclease V1 digestion, yet are unaffected by ribonuclease A, suggesting that much of the siRNA pool exists in double-stranded form.

We next tested whether P4R2 RNAs that accumulate in *dcl2/3/4* triple mutants are substrates for DCL3. In this experiment, RNA isolated from *dcl2/3/4* triple mutant plants was incubated with FLAG-tagged DCL3 that had been affinity purified from transgenic plants using anti-FLAG resin. In parallel control reactions, *dcl2/3/4* RNA was incubated with anti-FLAG resin that had been incubated with a cell-free lysate of non-transgenic plants (ecotype Col-0), or with anti-FLAG resin in reaction buffer with no added proteins (*Figure 2C*). Incubation of RNA from *dcl2/3/4* plants with affinity-purified

**Table 1.** RNA sequencing statistics.

| Sample | Yield (Mb) | %PF | Cluster (PF) | Q30 | Mean qual. (PF) |
|---|---|---|---|---|---|
| Col-0 (Rep 1) | 2223 | 88.24 | 22,229,855 | 87.65 | 34.24 |
| dcl2/3/4 (Rep 1) | 2611 | 88.03 | 26,111,516 | 87.11 | 34.03 |
| nrpd1-3 (Rep 1) | 3220 | 89.06 | 32,204,710 | 89.20 | 34.75 |
| rdr2-1 (Rep 1) | 3500 | 88.60 | 34,995,990 | 87.89 | 34.33 |
| Col-0 (Rep 2) | 3345 | 88.36 | 33,446,115 | 87.87 | 34.32 |
| dcl2/3/4 (Rep 2) | 3029 | 88.27 | 30,291,115 | 88.49 | 34.50 |
| nrpd1-3 (Rep 2) | 3014 | 88.49 | 30,141,955 | 88.63 | 34.54 |
| rdr2-1 (Rep 2) | 3301 | 88.05 | 33,005,785 | 87.20 | 34.08 |

FLAG-DCL3 resulted in the digestion of the >25 nt P4R2 RNAs into 24 nt RNA products (*Figure 2C*). By contrast, no digestion of the longer RNAs was observed in either control reaction. Collectively, the observations that P4R2 RNAs hybridize to probes representing both DNA strands, are sensitive to dsRNA-specific ribonuclease, and are processed into 24 nt RNAs by exogenous DCL3 supports the conclusion that P4R2 RNAs are the immediate precursors of 24 nt siRNAs in vivo.

## Sequencing of P4R2 RNAs

To identify RNAs that accumulate when dicing is abrogated, we gel-purified ~15–90 nt RNAs from wild-type (Col-0), *dcl2/3/4, pol IV (nrpd1* subunit null) or *rdr2* mutants and conducted RNA sequencing (RNA-seq) analyses using Illumina deep sequencing technology, performing two biological replicates for each sample (see *Tables 1 and 2* for sequencing and mapping statistics). In wild-type plants (ecotype Col-0), 24 and 23 nt siRNAs represent the most abundant small RNA class, as expected, outnumbering unique 21 nt small RNA species, which include miRNAs as well as siRNAs, by approximately 10-fold (*Figure 3*). In *dcl2/3/4* triple mutants, the small RNA profile is substantially altered. Because DCL1-dependent 21 nt RNAs are unaffected, they become the most abundant small RNA class in *dcl2/3/4* mutants. The dramatic loss of 23 and 24 nt siRNAs in *dcl2/3/4* triple mutants coincides with the occurrence of longer RNAs, mostly smaller than 45 nucleotides and displaying a peak at ~30 nt (see also *Figure 3*). These genome-wide RNA-seq results correlate closely with the RNA blot hybridization results observed in *Figure 2* at 5S rRNA gene loci. Indeed, a browser display of RNA-seq results focused on the region that corresponds to the 5S probes used for *Figure 2* (siR1003) shows that RNAs that accumulate in *dcl2/3/4* mutants range in size from 26–42 nt, with a mean size of 30 nt (*Figure 4*). These >25 nt RNAs represent both DNA strands and overlap extensively with 24 and 23 nt siRNAs. We conclude that the >25 nt RNAs that accumulate in *dcl2/3/4* triple mutants represent one or both strands of P4R2 duplex RNAs.

Although P4R2 RNAs accumulate to their highest levels in *dcl2/3/4* triple mutants, as visualized in the RNA blots of *Figure 2*, they are also detected by RNA-seq in wild-type plants. *Figure 5* shows

**Table 2.** Mapping statistics.

| Sample | Total | Perfect match | % | Single mismatch | % |
|---|---|---|---|---|---|
| Col-0 (Rep 1) | 9,742,599 | 8,602,123 | 88.29 | 1,140,476 | 11.71 |
| dcl2/3/4 (Rep 1) | 8,440,663 | 7,179,444 | 85.06 | 1,261,219 | 14.94 |
| nrpd1-3 (Rep 1) | 8,966,872 | 7,799,160 | 86.98 | 1,167,712 | 13.02 |
| rdr2-1 (Rep 1) | 9,261,683 | 8,081,911 | 87.26 | 1,179,772 | 12.74 |
| Col-0 (Rep 2) | 13,955,193 | 12,179,431 | 87.28 | 1,775,762 | 12.72 |
| dcl2/3/4 (Rep 2) | 10,119,912 | 8,570,313 | 84.69 | 1,549,599 | 15.31 |
| nrpd1-3 (Rep 2) | 10,285,064 | 9,096,528 | 88.44 | 1,188,536 | 11.56 |
| rdr2-1 (Rep 2) | 9,970,701 | 8,793,492 | 88.19 | 1,177,209 | 11.81 |

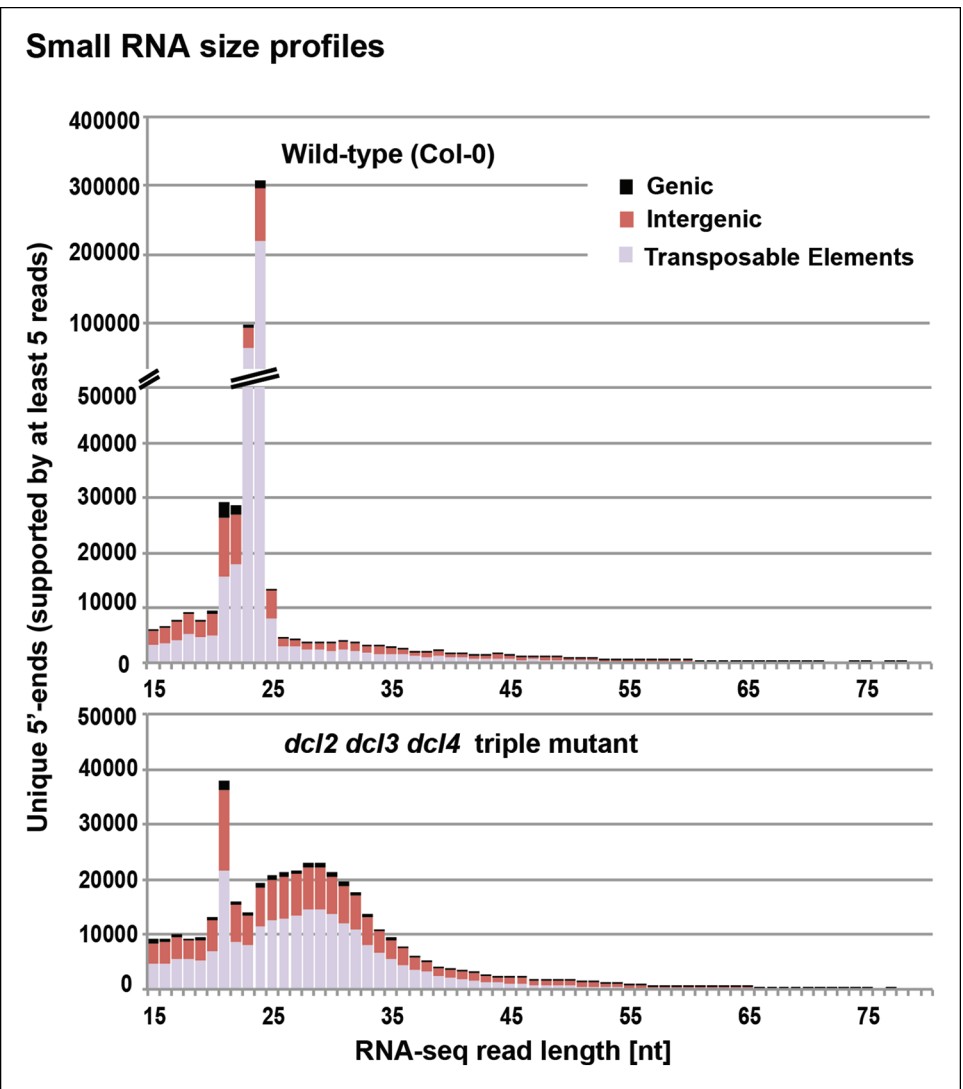

**Figure 3.** Small RNA size profiles in wild-type and dcl2 dcl3 dcl4 (*dcl2/3/4*) triple mutants. The number of unique sequences starting at a given 5' terminal nucleotide (and represented by at least five reads after normalization and filtering) is plotted for RNAs in each size class. Color coding in each bar depicts the relative proportions of RNA sequencing (RNA-seq) reads that correspond to genes, intergenic regions, or transposable elements. *Figure 3—figure supplement 1* provides related data, showing the total numbers of unique 15–94 nt RNA sequences (one or more copies) detected in wild-type or *dcl2/3/4* mutants.

The following figure supplement is available for figure 3:

**Figure supplement 1.** Number of unique sequences among RNAs of 15–94 nt in wild-type or dcl2 dcl3 dcl4 (*dcl2/3/4/*) triple mutants.

browser views of three representative loci selected by virtue of having five or more identical reads for at least one P4R2 RNA mapping to the locus in the wild type (Col-0). Browser views of three additional loci are shown in *Figure 5—figure supplement 1*. In each case, P4R2 RNAs whose positions correspond closely to those of abundant 23 and 24 nt siRNAs (only 24 nt siRNAs are shown) are readily detected in dcl2/3/4 mutants and in Col-0, but are absent in pol IV or rdr2 null mutants, confirming a requirement for both Pol IV and RDR2 for P4R2 RNA biogenesis. Genome-wide, at least 5500 genomic intervals to which 24 nt siRNAs map have overlapping 26-94 nt P4R2 RNAs that are detectable in Col-0, and typically increase in abundance in dcl2/3/4 mutants, but are absent, or substantially reduced, in pol IV or rdr2 null mutants in one or both biological replicates

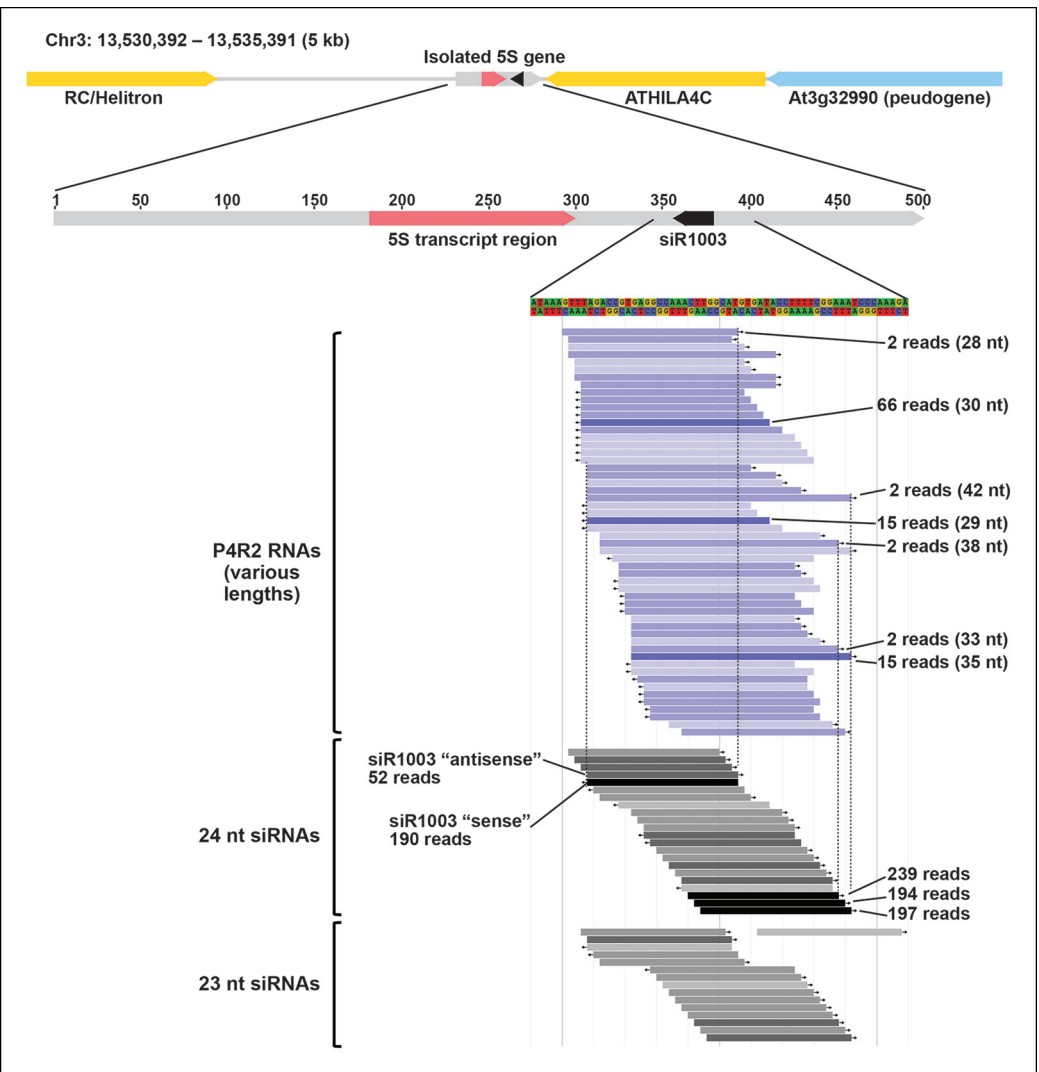

**Figure 4.** Browser view of Pol IV/RDR2-dependent RNAs (P4R2 RNAs) and 24/23 nt small interfering RNA (siRNAs) in the intergenic spacer region of a 5S ribosomal RNA (rRNA) gene repeat unit. An isolated 5S rRNA gene repeat (~500 bp, gray horizontal bar with red transcript region) is shown within its 5 kb chromosomal context, flanked by two transposable elements, shown in yellow, and a pseudogene, shown in blue. Below the diagram, P4R2 RNAs are depicted as horizontal bars shown in shades of blue whereas 24 and 23 nt siRNAs are shown in shades of gray to black, with color intensity reflecting abundance (read counts are provided for several examples). Each bar represents a specific RNA sequence, with arrows depicting the RNA strand orientation relative to the reference genome sequence (TAIR10). Dotted vertical lines provide alignments and show that the ends of highly abundant (>100 reads) siRNA species tend to coincide with the ends of P4R2 RNAs for which there is more than a single read.

(*Supplementary file 1*). Importantly, the number of mapped RNA-seq reads was similar for the Col-0, dcl2/3/4, pol IV and rdr2 RNA-seq datasets (see *Table 2*).

Consistent with the genome browser views of *Figures 4 and 5*, P4R2 RNAs that accumulate in *dcl2/3/4* triple mutants show a high degree of correspondence to 24 and 23 nt siRNA sequences, genome-wide. To simplify the discussion we focus solely on 24 nt siRNAs. Ninety-six percent of P4R2 RNAs identified in *dcl2/3/4* mutants fully encompass or overlap 24 nt siRNA sequences (Figure 6A). Among the P4R2 loci we identified are 20 of the 22 loci that Li et al. (*Li et al., 2015*) confirmed by reverse transcription-polymerase chain reaction to be capable of generating Pol IV and RDR2-dependent RNAs (*Supplementary file 2*).

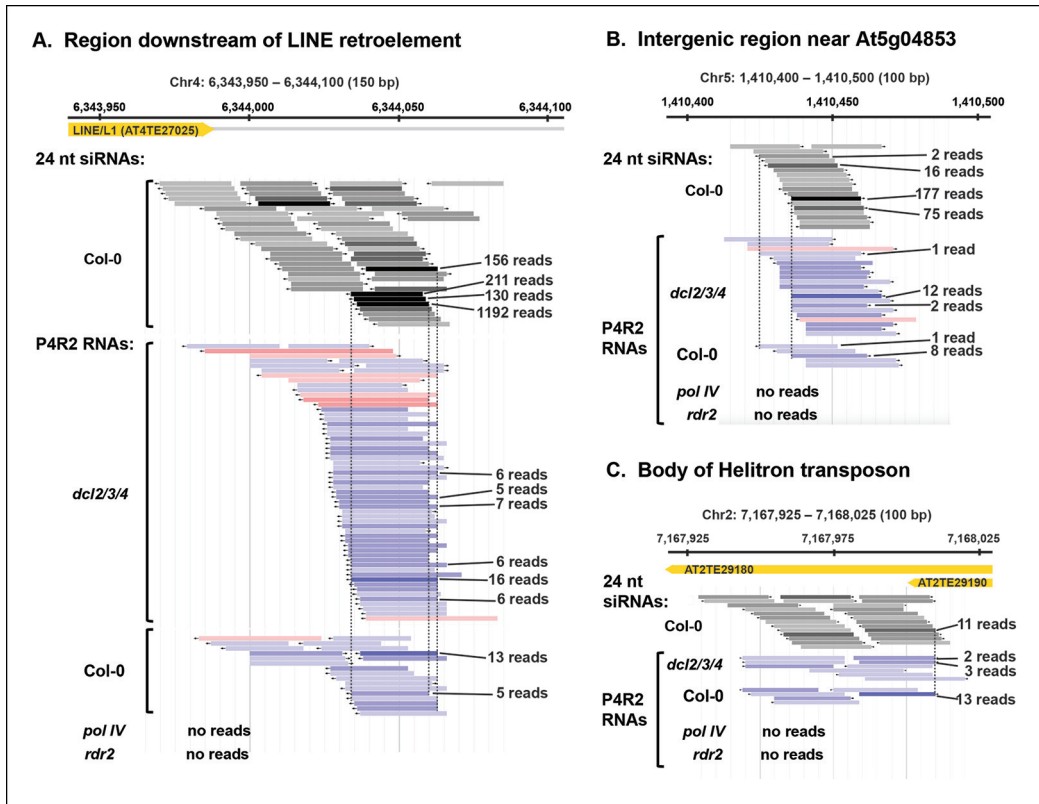

**Figure 5.** Pol IV/RDR2-dependent RNAs (P4R2 RNAs) are dependent on both Pol IV and RDR2. (**A–C**)Browser views of three 24 nt small interfering RNA (siRNA) loci at which abundant P4R2 RNAs that accumulate in dcl2 dcl3 dcl4 (*dcl2/3/4*) mutants are also observed in wild-type plants (Col-0) but not in *nrpd1* (Pol IV) or *rdr2* mutants. These examples are representative of a subset of loci selected by virtue of having five or more reads for at least one of the P4R2 species at the locus in wild-type (Col-0) plants. These loci tend to correspond to loci giving rise to abundant siRNAs. Vertical dotted lines provide alignments between abundant siRNAs and the P4R2 RNAs. P4R2 RNAs that are 40 nt or longer are shown in shades of pink. RNAs of 25-39 nt are shown in shades of blue. Browser views for three additional representative loci are shown in *Figure 5—figure supplement 1* . *Supplementary file 1* provides a table with coordinates for thousands of 100 bp genomic intervals in which 24 nt siRNA loci and putative P4R2 RNAs are detected in wild-type and *dcl2/3/4*, but are depleted in *nrpd1* and *rdr2* mutants.

The following figure supplement is available for figure 5:

**Figure supplement 1.** P4R2 RNAs are co-dependent on Pol IV and RDR2.

We next examined how P4R2 RNAs and their matching siRNAs align with one another. These analyses revealed that siRNAs typically correspond to either the extreme 5' end or 3' end of a P4R2 RNA (*Figure 6B*), with each of these scenarios represented by ~100,000 examples (see the peaks at offset position zero). This suggests that 24 nt siRNAs are most frequently generated by single internal dicing events, with the cut site measured from either end of a P4R2 precursor duplex. The slight skewing (non-symmetry beneath the peak) of the alignment profiles in *Figure 6B* indicates that siRNAs whose ends do not precisely coincide with the ends of P4R2 RNAs tend to be internal to P4R2 RNAs.

## Consensus sequences of P4R2 RNAs and siRNAs

To further explore the observation that siRNAs are preferentially derived from the 5' or 3' ends of P4R2 RNAs, we searched for sequence motifs that might be common to the ends of P4R2 RNAs and siRNAs. Most 24 nt siRNAs begin with an adenosine (A) at the 5' end (*Figure 7A*), which has been thought to reflect the RNA binding specificity of AGO4 (*Mi et al., 2008*). By contrast, P4R2 RNAs show a strong preference for having a purine at their 5' ends, with  A or G present at similar

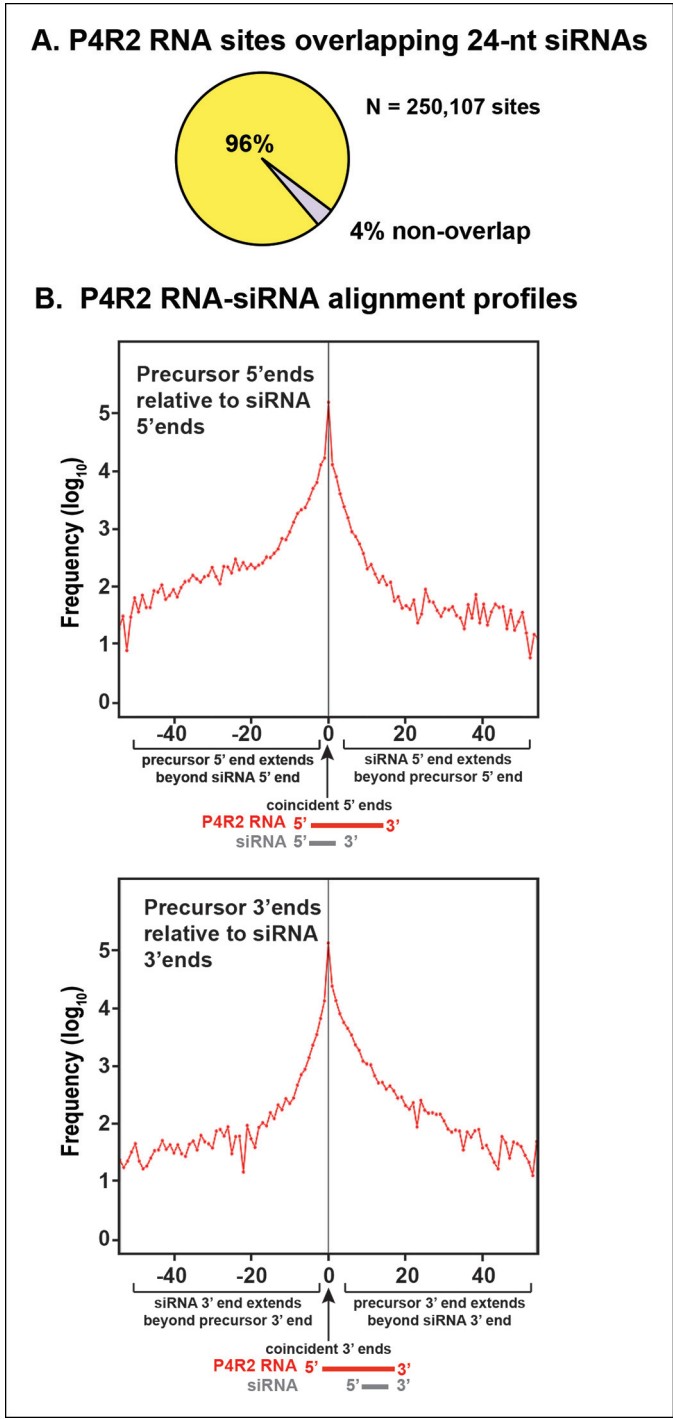

**Figure 6.** Sequence relationships between Pol IV/RDR2-dependent RNAs (P4R2 RNAs) and small interfering RNA (siRNAs). (**A**) Correspondence between P4R2 RNA and siRNA loci. P4R2 RNAs were mapped to the Arabidopsis reference genome (TAIR10) and the frequency at which 24 nt siRNAs overlap these P4R2 genomic positions was calculated. To be considered for this analysis, specific siRNAs had to be represented by at least five reads in wild-type Col-0. *Supplementary file 2* shows that P4R2 RNA loci include loci confirmed by *Li et al., 2015* to generate Pol IV-dependent transcripts. (**B**) P4R2 RNA and 24 nt siRNA spatial relationships. The top panel shows the frequency distribution of P4R2 RNA 5′ end positions relative to siRNA 5′ ends. At position zero on the x-axis, P4R2 RNAs and siRNAs share the same 5′ terminus. Negative values indicate how far (in nucleotides) the 5′ end of the P4R2 RNA is located upstream of an siRNA start position. Likewise, positive values indicate how far the 5′ end of a P4R2 RNA is located downstream of an siRNA start position. The lower panel shows the frequency with which P4R2 RNAs and siRNAs align at 3′ ends. At position zero on the x-axis, P4R2 RNAs and siRNAs share the

*Figure 6. continued on next page*

*Figure 6. Continued*

same 3' terminus. Negative values occur when P4R2 RNAs end upstream of siRNA 3' ends, and positive values occur when P4R2 RNAs end downstream of siRNA 3' ends (computed using FEATnotator, v1.2.2, *Podicheti and Mockaitis, 2015*).

frequency (*Figure 7A*). Aside from this difference, the 5' ends of P4R2 RNAs and 24 nt siRNAs display intriguing similarities. In both cases, the purine at the 5' terminus is preferentially preceded by a T in the corresponding genomic sequence, and is preferentially followed by an A or U at the second position (*Figure 7B*, compare top two sequence logos).

At their 3' ends, 24 nt siRNAs preferentially end with a uracil (U), with the order of nucleotide preference at this 3' terminal position being U > C > A. P4R2 RNAs also tend to end with U, but the order of nucleotide preference at their 3' termini is slightly different: U > A > C. Interestingly, a larger (but weak) consensus, ACU can be discerned at the 3' end of P4R2 RNAs (*Figure 7B*; compare top two sequence logos). These 5' and 3' consensus sequences, determined for P4R2 RNAs considered *en masse*, regardless of length, hold true for P4R2 RNAs of each length considered separately (see *Figure 7—figure supplement 1*).

We next examined whether consensus sequences differ for P4R2 RNAs that begin with A versus those that begin with G at their 5' ends (*Figure 7B*; bottom two sequence logos). Interestingly, P4R2 RNAs that begin with A account for the weak 3' ACU consensus that is detected in analyses of total P4R2 RNAs. By contrast, P4R2 RNAs that begin with G lack the ACU signature, but instead show a preference for U > C > A at the 3' terminal position. Interestingly, U > C > A is the same nucleotide preference order observed at the 3' termini of 24 nt siRNAs. Collectively, these observations suggest that P4R2 RNAs that begin with A might tend to give rise to 24 nt siRNAs from their 5' ends, whereas P4R2 RNAs that begin with G might tend to give rise to 24 nt siRNAs from their 3' ends. It is noteworthy that P4R2 RNAs that begin with G show a strong tendency to end with U, such that the complementary strand (which can be considered the bottom strand) is expected to begin with A. Thus, it may be that DCL3 cutting of precursor duplexes preferentially takes place at a fixed distance from a 5' terminal adenosine present on either strand of the duplex. This hypothesis is consistent with in vitro data of Nagano et al. demonstrating that DCL3 preferentially cleaves dsRNAs with A or U at the 5' end (*Nagano et al., 2014*).

## Pol IV transcripts synthesized in vitro resemble P4R2 RNAs

Pol IV, like Pol II, will initiate transcription from single-stranded oligonucleotide DNA templates in vitro (*Haag et al., 2012*). To assess the sizes of RNA transcripts that Pol IV is capable of making in vitro, as well as the possibility that Pol IV may initiate or terminate transcription with favored nucleotides, we used a 7249 nt single-stranded bacteriophage M13 genome as a DNA template and compared transcripts made by affinity-purified Pol IV or II (*Figure 8*). For this experiment, Pol IV and II were affinity purified by virtue of FLAG epitope tags fused to the C-termini of their NRPD1 or NRPB2 subunits, respectively (*Haag et al., 2012*; *Pontes et al., 2006*; *Ream et al., 2009*). Previous immunoblot and mass spectrometry studies have shown that affinity-purified Pols II and IV are free of contaminating RNA polymerases (*Haag et al., 2012*; *Ream et al., 2009*). In the case of FLAG-tagged Pol IV, the NRPD1-FLAG transgene was expressed in an *nrpd1 rdr2* null mutant background (*Haag et al., 2012*), precluding any potentially confounding results due to transcription by associated RDR2.

RNA-seq analyses revealed that Pol IV and II each initiate transcription in a mostly unbiased manner with respect to position on the circular M13 genome, with Pol IV and II each initiating at more than 2000 positions. The size profiles of RNAs synthesized by Pol IV or II differ, with Pol IV transcripts having mean and median lengths of 44 and 37 nt, respectively, and Pol II transcripts having mean and median lengths of 85 and 68 nt (*Figure 8A,B*). The 5' ends of Pol IV and II transcripts are similar, each initiating with a purine just downstream of a pyrimidine, as is typical of DNA-dependent RNA polymerases (*Figures 8C,D*), but with Pol IV transcripts also displaying a tendency to have A or G at the +2 RNA position (*Figures 8C*). Interestingly, Pol IV transcripts in vitro tend to end with a C or U (T in the DNA), reminiscent of the tendency to have a U or C (but also A) at the 3' ends of P4R2

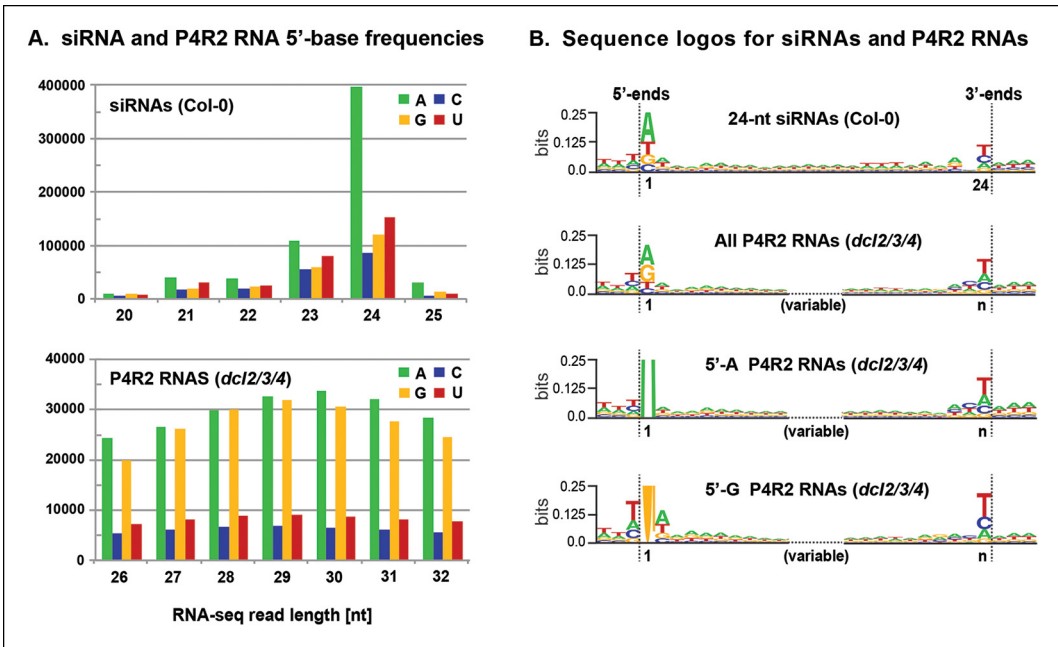

**Figure 7.** Sequence features of 24 nt small interfering RNAs (siRNAs) and Pol IV/RDR2-dependent RNA (P4R2 RNA) precursors. (**A**) Frequencies at which the four nucleotides are present at the 5′ terminus of 20–25 nt small RNAs in wild-type (Col-0) plants or at the 5′ termini of P4R2 RNAs in *dcl2 dcl3 dcl4* triple mutants (*dcl2/3/4*). Data for the subset of P4R2 RNAs in the peak size range of 26–32 nt is shown. (**B**) Sequence logos for chromosomal DNA sequences corresponding to all 24 nt siRNAs (in wild-type, Col-0), all P4R2 RNAs, P4R2 RNAs that begin with adenosine (5′-A) or P4R2 RNAs that begin with guanosine (5′-G). The P4R2 logos were generated using *dcl2/3/4* triple mutant data. The logos include three nucleotides upstream and three nucleotides downstream of the chromosomal DNA sequences that match the RNAs. Each unique RNA sequence is represented only once in the input data. For reads mapping to multiple loci, upstream and downstream DNA sequences were obtained from one mapped site selected at random. Graphics were generated using WebLogo v2.8.2 (*Crooks et al., 2004*). *Figure 7—figure supplement 1* provides sequence logos for P4R2 RNAs of varying length, showing that consensus sequences are consistent among these RNAs.

The following figure supplement is available for figure 7:

**Figure supplement 1.** Sequence logos for Pol IV/RDR2-dependent RNAs (P4R2 RNAs) of different lengths.

RNAs in vivo. But unlike P4R2 RNAs in vivo, Pol IV in vitro transcripts have a very strong preference for an A at the template position immediately downstream of the RNA 3′ end (*Figure 8C*).

## Unexpected 3′ terminal nucleotides are sometimes found in P4R2 RNAs

A genome browser view of P4R2 precursor RNAs aligned with 24 nt siRNAs at an *AtSN1* retrotransposon located on chromosome 3, a target of RNA-directed DNA methylation we have examined in numerous previous studies (*Blevins et al., 2014*; *Haag et al., 2009*; *Onodera et al., 2005*; *Ream et al., 2009*; *Tan et al., 2012*; *Wierzbicki et al., 2008, 2009*), is shown in *Figure 9A*. In this case, RNA species that differ by one nucleotide are displayed separately, whereas in previous browser views (*Figures 4 and 5*), single-nucleotide mismatches were tolerated. Mismatched nucleotides are color coded red (U; T in the context of DNA), green (A), tan (G) or blue (C) in *Figure 9A*. As is readily apparent, the 3′ terminal nucleotide of P4R2 RNAs and 24 nt siRNAs sometimes does not match the expected sequence. Approximately 5% of all P4R2 RNAs have such 3′ mismatched nucleotides. A graphical depiction of mismatch frequency at each nucleotide position for RNAs of discrete lengths, from 15 nt (top) to 76 nt (bottom), is shown in *Figure 9B*. To generate this plot, background mismatch/error levels (% of total reads containing a mismatch) observed at each position for each RNA size class in RNAs of wild-type plants were subtracted from the values obtained for RNAs of the same length in *dcl2/3/4* triple mutants. The difference was then plotted, with the

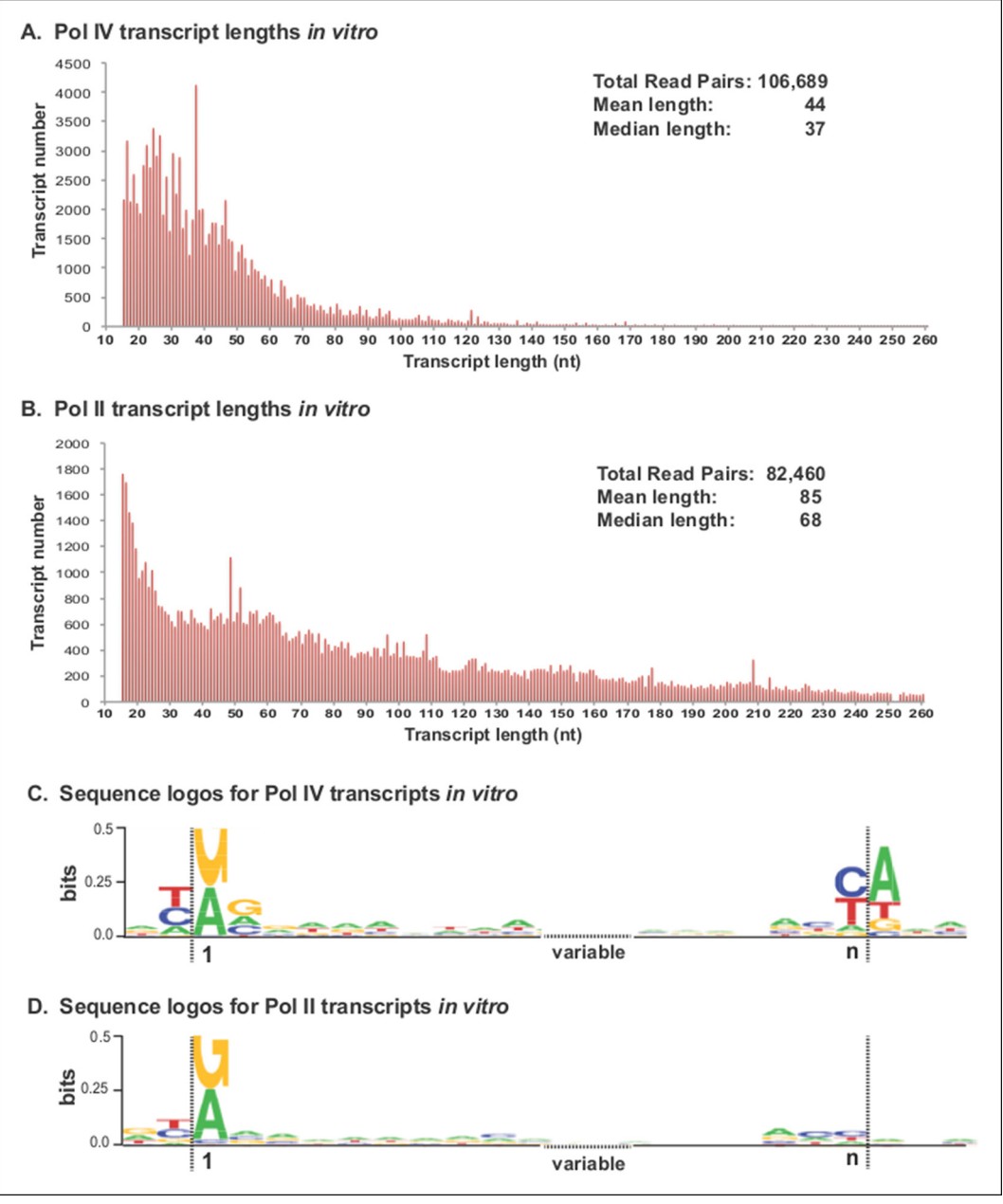

**Figure 8.** Pol IV transcripts generated in vitro share features of Pol IV/RDR2-dependent RNAs (P4R2 RNAs) in vivo. (**A** and **B**) Size and frequency of RNAs transcribed by polymerase (Pol) IV or II in vitro. Pol IV and Pol II were affinity purified by virtue of FLAG epitope tags fused to the C-termini of the NRPD1 or NRPB2 subunits, respectively. In the case of Pol IV, the transgenic *NRPD1-FLAG* line is null for the endogenous *NRPD1* and *RDR2* genes, such that Pol IV is free of associated RDR2. Transcripts generated using closed-circular single-stranded M13 virus as the DNA template were subjected to RNA-seq. The frequency and sizes of mapped reads are plotted. (**C** and **D**) Sequence logos for the 5′ and 3′ ends of Pol IV and II in vitro transcripts. RNA-seq, RNA sequencing.

intensity of the color reflecting mismatch frequency. Only read sequences with single mismatches or perfect matches to the Arabidopsis genome were considered for this analysis. An enrichment of mismatched nucleotides at the 3′ terminal position of 24 nt RNAs and at the 3′ ends of RNAs longer than 26 nt is readily apparent, with the strongest signals observed among RNAs of 26–37 nt, closely matching the peak size distribution of P4R2 RNAs.

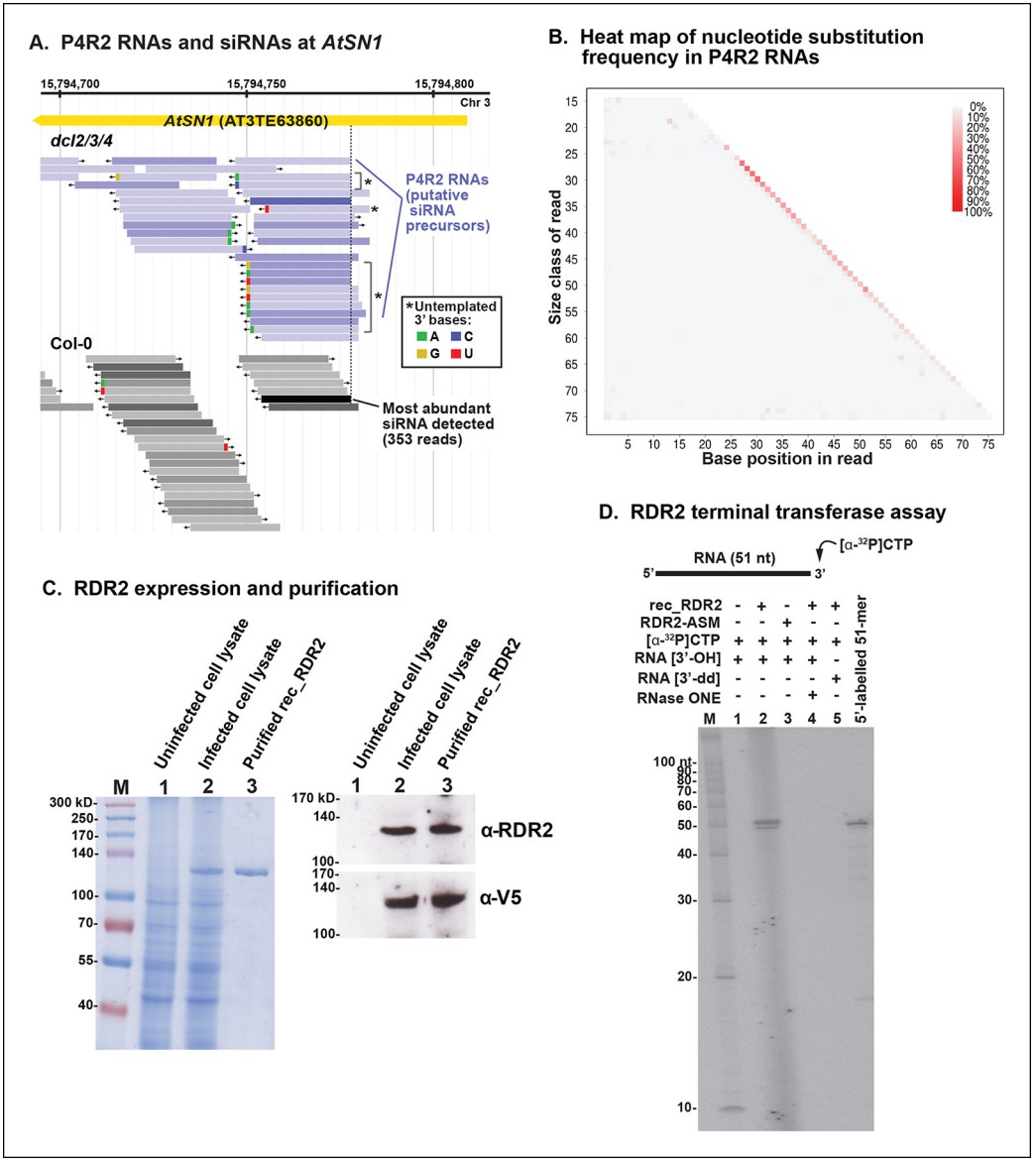

**Figure 9.** 3' mismatches detected in 24 nt siRNAs and Pol IV/RDR2-dependent RNAs (P4R2 RNAs) may reflect RDR2 terminal transferase activity. (**A**) Genome browser view of P4R2 RNAs (shades of blue) and 24 nt small interfering RNA (siRNAs) (shades of gray) at a representative locus, an *AtSN1* retrotransposon on chromosome 3. Each horizontal bar represents a specific RNA sequence (RNA-seq), with arrows depicting their direction relative to the Arabidopsis reference genome sequence (TAIR10). The intensity of shading reflects the abundance of each RNA species in the RNA-seq dataset. Brightly colored nucleotides, color coded for A, G, C, or U (see inset), represent nucleotides that do not match the corresponding DNA sequence of the locus. The dotted line highlights the coincident 5' ends of the most abundant P4R2 RNAs at the locus (colored deep purple) and the most abundant siRNAs (colored black). (**B**) Heat map depicting the frequency of mismatched nucleotides at each position of RNAs ranging in size from 15 to 76 nt in *dcl2 dcl3 dcl4* triple mutant plants. To correct for the frequency of errors inherent to sequencing, mismatch values for each position of 15–76 nt RNAs in wild-type plants were subtracted prior to plotting the data. Only read sequences with single mismatches or perfect matches to the reference genome were utilized for this analysis. (**C**) Over-expression and purification of recombinant RDR2. The image on the left shows a 7.5% sodium dodecyl sulfate-polyacrylamide gel electrophoresis(SDS-PAGE) gel, stained with Coomassie blue, showing molecular weight markers (M), proteins of un-infected High Five cells (lane 1), proteins of High Five cells 72 hr after infection with baculovirus expressing recombinant RNA-dependent RNA polymerase 2 (RDR2) (lane 2), and purified recombinant V5-tagged RDR2 after affinity purification and elution with V5 peptide. The image at right shows anti-RDR2 and anti-V5 immunoblots of the same three protein samples. For RDR2 detection, rabbit anti-RDR2 primary antibody was used in conjunction with donkey anti-rabbit HRP-

*Figure 9. continued on next page*

*Figure 9. Continued*

conjugated secondary antibody. Detection of V5-tagged RDR2 involved anti-V5 HRP conjugate antibody. (**D**) RDR2 terminal transferase activity. Recombinant RDR2 or an active-site mutant form of RDR2 (RDR2-ASM) was incubated with alpha-labeled $^{32}$P-CTP and 51 nt RNA substrates bearing 3′ hydroxyl or 3′ dideoxy termini. Reaction products were subjected to denaturing polyacrylamide gel electrophoresis (PAGE) and autoradiography. For gel lane 4, reaction products were treated with RNase One, which degrades single-stranded RNAs, prior to PAGE. RNA size markers were run in lane M. The 51 nt RNA template, 5′ end-labeled using T4 polynucleotide kinase, was run as a size marker in the lane at far right.

## RDR2 has terminal transferase activity

We overproduced full-length recombinant RDR2 (recRDR2) by expressing the protein in insect cells using a baculovirus vector system (*Figure 9C*). As shown in *Figure 9D*, RDR2 has terminal transferase activity. In the experiment shown, a 51 nt RNA was incubated with recRDR2 and radioactive, alpha-labeled $^{32}$P-CTP, but no other nucleoside triphosphates. RDR2 catalyzes the addition of one or two non-templated radioactive cytosines to the 3′ end of the RNA (*Figure 9D*, lane 2). This 3′ end-labeling fails to occur using an active site mutant version of RDR2 (RDR2-ASM), in which the magnesium-binding aspartates of the catalytic center have been changed to alanines (lane 3). Labeling also fails to occur if the 3′ terminal nucleotide of the RNA substrate lacks a 3′ hydroxyl group on the ribose, but instead has a 2′, 3′-dideoxy ribose (lane 5). The end-label added by RDR2 is sensitive to RNase 1, indicating that labeling occurs in the context of single-stranded RNA (lane 4). Analogous terminal transferase activity has not been detected in Pol IV fractions. Collectively, these results suggest that the 3′ mismatches observed at low frequency among P4R2 RNAs might be telltale signs of RDR2 terminal transferase activity, potentially acting on completed Pol IV transcripts, RDR2 transcripts, or both.

## Discussion

A recent study reported the identification of Pol IV and RDR2-dependent RNAs whose characteristics included a 5′-monophosphate group, a correspondence to known 24 nt siRNA-producing loci and accumulation in *dcl2/3/4* triple mutants (*Li et al., 2015*). These transcripts were deduced to be hundreds of nucleotides in length, on average, based on the computational joining of overlapping RNA-seq reads (*Li et al., 2015*). In general, the P4R2 RNAs we identified have similar properties to the RNAs defined by Li et al., with 20 of the 22 loci scrutinized most carefully by Li et al. matching P4R2 RNAs from our study. Thus, our results extend the results of Li et al., but indicate that the vast majority of Pol IV and RDR2-dependent RNAs are smaller than 45 nt (see *Figure 3*). Dozens of P4R2 RNAs can overlap at a given locus, as shown in *Figures 4 and 5*, but we conclude that individual P4R2 RNAs represent individual Pol IV-RDR2 transcription units.

Our data indicate that the short double-stranded P4R2 RNAs that accumulate in *dcl3* mutants are the immediate precursors of siRNAs, based on the ability of DCL3 to cut these precursors into 24 nt RNAs in vitro. These results are consistent with those of Nagano et al., who showed that DCL3 preferentially cleaves short synthetic dsRNAs of 30–50 nt, but not longer RNAs (*Nagano et al., 2014*). Our analyses indicate that 24 nt siRNAs are preferentially generated from the 5′ or 3′ ends of P4R2 RNAs, suggesting that P4R2 duplex RNAs can be diced from either end, also in agreement with Nagano et al.'s studies. Generation of siRNAs from the ends of P4R2 RNAs, rather than from internal sequences, leads to the prediction that each P4R2 RNA sequence has limited capacity to specify multiple alternative siRNAs. Thus, the multitude of unique 24 nt siRNA sequences present in wild-type cells can be expected to be derived from a similarly large number of P4R2 precursors, depending on whether one or both strands of diced duplexes are utilized as siRNAs. Consistent with this prediction, our sequenced datasets included 1,055,471 unique 23 or 24 nt siRNA sequences (in wild-type plants) and 933,453 unique P4R2 RNAs of 26–94 nt (in *dcl2/3/4* triple mutants).

An expectation is that if Pol IV generates single-stranded RNAs that serve as templates for RDR2, these transcripts would accumulate in an *rdr2* mutant and be detectable on RNA blots using sense or antisense probes that detect abundant siRNAs (such as the 5S gene intergenic spacer siRNAs). Alternatively, if RDR2 were to act upstream of Pol IV, which is contrary to current models, but a

formal possibility, one might similarly expect to detect RDR2 transcripts in a *pol IV* mutant. However, our RNA blot results (*Figure 2*) defy these predictions, as do the RNA-seq data for *pol IV* and *rdr2* mutants. Instead, precursors are depleted if either RDR2 or Pol IV is absent. This suggests that Pol IV and RDR2 are each required for the synthesis of both strands of P4R2 RNA duplexes, as was also noted by Li et al. (*Li et al., 2015*), which fits with the physical association of Pol IV and RDR2 and the hypothesis that the two enzymes function as a complex (*Haag et al., 2012*; *Law et al., 2011*). We have shown previously that RDR2 activity detected in vitro using affinity-purified Pol IV–RDR2 complexes from wild-type plants is not apparent using RDR2 affinity purified from a *pol IV* mutant (*Haag et al., 2012*). Although Pol IV isolated from an *rdr2* null mutant is active in vitro (*Figure 8*; see also [*Haag et al., 2012*]), it is possible that Pol IV may require RDR2 association for activity in vivo.

An important ramification of Pol IV and RDR2 co-dependence is that it is unclear if the P4R2 RNAs we cloned and sequenced in our study represent Pol IV transcripts, RDR2 transcripts, or both. Although we detect P4R2 RNAs corresponding to both strands at many (but not all) loci, as can be seen in the browser displays of *Figures 4, 5 and 9A*, these could be transcripts of only Pol IV or, conversely, only RDR2. Sequences enriched at the ends of P4R2 RNAs preclude easy assignment of which polymerase generated them. For instance, P4R2 RNAs tend to begin with a purine and end with a pyrimidine such that the complementary strands should also be enriched for purines at the 5' end and pyrimidines at the 3' end. The terminal transferase activity inherent to RDR2 provides a plausible explanation for mismatched, untemplated nucleotides at the 3' ends of some (~5%) P4R2 RNAs, but this activity could modify completed RNA transcripts made by Pol IV, RDR2, or both. Despite these uncertainties, P4R2 RNAs preferentially begin with a purine that, in the context of the

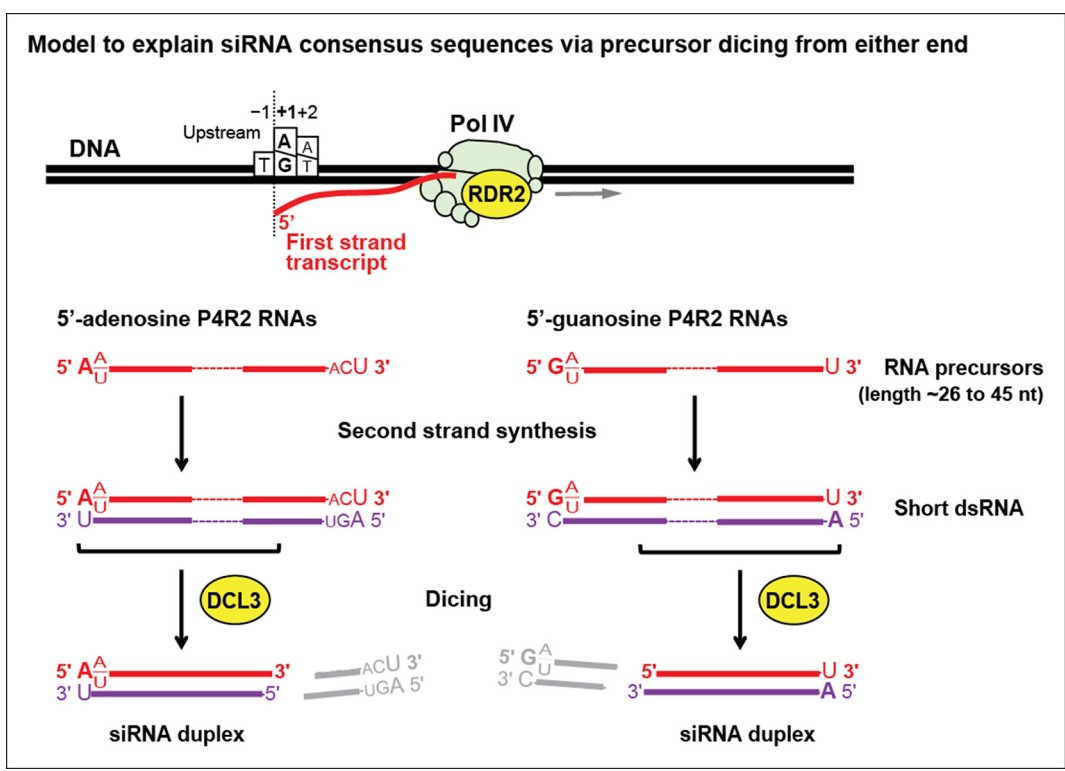

**Figure 10.** Model for the biogenesis of 24 nt small interfering RNAs (siRNAs) via single dicing events. Pol IV/RDR2-dependent RNAs (P4R2 RNAs) tend to begin with a purine (A or G) at position 1, adjacent to what would be a T at position −1 in the corresponding DNA strand and an A (or U) at position 2. A similar signature is detected at the 5' end of 24 nt siRNAs (see *Figure 7*), except that only A (not G) is enriched at the 5' terminus of these siRNAs. Thus, DICER-like 3 (DCL3) cleavage measured from the 5' adenosine of those P4R2 RNAs that begin with A could account for the similar 5' end sequences of P4R2 RNAs and 24 nt siRNAs. P4R2 RNAs that begin with A tend to end with ACU, which is not reflected in siRNAs. However, P4R2 RNAs that begin with G tend to end with a 3' U, with U > C > A being the order of preference, which matches the 3' end consensus for 24 nt siRNAs. Because P4R2 RNAs that begin with G most often end with U, their complementary strands (purple bottom strands in the figure) would tend to have A at their 5' ends. DCL3 dicing, measured from this 5' A, would liberate a top strand (red in the figure) whose 3' end could account for the 3' consensus (U > C > A) of 24 nt siRNAs.

genome sequence, tends to occur immediately downstream of a pyrimidine (in the equivalent DNA strand). This pyrimidine–purine rule is a characteristic of DNA-dependent RNA polymerases, in general, and has a structural basis that involves base-stacking interactions between the initiating (+1) RNA nucleotide (a purine) and −1 DNA template nucleotide (also a purine), in addition to the base-pairing that occurs between the +1 RNA and +1 DNA template bases (*Basu et al., 2014*). The pyrimidine–purine rule has been shown to apply to protein-coding genes throughout the Arabidopsis genome (*Cumbie et al., 2015*), to Arabidopsis pre-rRNAs transcribed by multisubunit RNA Pol I (these initiate with an A downstream of a T; *Doelling et al., 1993*) and to 5S genes transcribed by RNA polymerase III (initiating with a G downstream of a T). Our in vitro transcription studies using affinity-purified Pol IV or Pol II also recapitulate the pyrimidine–purine rule. Moreover, Pol IV generates relatively short RNAs in vitro, with size distributions and 3′ ends resembling those of P4R2 RNAs. Collectively, these observations suggest that P4R2 RNA biogenesis is initiated by Pol IV, as depicted in *Figure 10*, with limited processivity of Pol IV accounting for the relatively short size of P4R2 RNAs. In our model, designed to account for the consensus sequences of P4R2 RNAs and 24 nt siRNAs (refer to *Figure 7*), Pol IV generates first strands that initiate with a purine. Those RNAs that initiate with A best account for siRNA 5′ end sequences, whereas RNAs that initiate with G best account for siRNA 3′ end sequences, suggesting that resulting duplexes, following second strand synthesis by RDR2, are diced from opposite ends. Because first strand RNAs that begin with G tend to end with U, their complementary strands would tend to begin with A. Thus dicing measured from a 5′ adenosine might be a preference underlying the alternative dicing of P4R2 duplexes from either end.

## Materials and methods

### Plant materials

The *A. thaliana* mutants *rdr2-1*, *nrpd1-3 (pol IV)* and *nrpe1-11 (pol V)* were described in *Xie et al., 2004*; *Onodera et al., 2005*; and *Pontes et al., 2006* respectively. A line homozygous for *ago4-7* was selected from seed obtained from the Arabidopsis Biological Resource Center (stock ID: SALK_095100). The single mutants *dcl2-5, dcl3-1, and dcl4-2*; double mutants *dcl2-5 dcl3-1 (dcl2/3)*, *dcl2-5 dcl4-2 (dcl2/4)* and *dcl3-1 dcl4-2 (dcl3/4)*; and the triple mutant *dcl2-5 dcl3-1 dcl4-2 (dcl2/3/4)* were described in (*Blevins et al., 2006*). Genetic crosses between *dcl2/3/4* and above-mentioned single mutants were performed to generate the quadruple mutants: *rdr2-1 dcl2-5 dcl3-1 dcl4-2*, *nrpd1-3 dcl2-5 dcl3-1 dcl4-2*, *nrpe1-11 dcl2-5 dcl3-1 dcl4-2* and *ago4-7 dcl2-5 dcl3-1 dcl4-2*. A transgenic plant line in which the *dcl3-1* mutant was complemented by a full-length DCL3 genomic clone engineered to possess a FLAG epitope tag at the C terminus (pEarleyGate 302-gDCL3-FLAG) was previously described (*Pontes et al. 2006*).

### RNA blot hybridization

Total RNA (~100 µg) was extracted from leaves of 2-week-old plants using TRIzol (ThermoFisher Scientific, Grand Island, NY) and then size-fractionated on RNeasy Mini Columns (Qiagen, Valencia, CA) as in *Blevins et al. (2006)*. The low-molecular-weight RNA fraction was then separated on a 16% polyacrylamide gel and transferred to a nylon membrane by electroblotting. Different $^{32}$P-end-labeled DNA oligonucleotides were used for successive hybridizations: the probe for 'siR1003 sense' was 5′-ATG CCT ATG TTG GCC TCA CGG TCT-3′, for 'siR1003 antisense' was 5′- AGA CCG TGA GGC CAA CAT AGG CAT-3′, for 'ta-siR255' was 5′-TAC GCT ATG TTG GAC TTA GAA-3′ and for 'miR160' was 5′-TGG CAT ACA GGG AGC CAG GCA-3′. Size standards were synthetic RNA oligonucleotides of 15, 21, 24, 30, and 50 nt lengths. Three- to five-day exposures of the blot were scanned on a Typhoon 9210 phosphorimager system (GE Healthcare, Pittsburgh, PA).

### RNA sequencing

Custom libraries for ~15–90 nt RNAs were constructed and sequenced by Fasteris SA (http://www.fasteris.com/). In brief, 5 µg total RNA was subjected to 15% denaturing polyacrylamide gel electrophoresis. The desired RNA size-range was excised and eluted, and 3′ and 5′ adapters were added according to the ligation-based Illumina small RNA-seq protocol. Sequencing by synthesis (1 × 100) was performed on an Illumina HiSeq 2500 (Illumina, San Diego, CA) instrument following

manufacturer's recommendations (TruSeq SBS Kit v3; Illumina, San Diego, CA). Basecalling was performed using the HiSeq Control Software (v2.0.10.0), RTA (v1.17.21.3) and CASAVA (v1.8.2). Insert lengths ranged from 15-94 nt following adapter sequence removal. RNA-seq statistics are provided in *Table 1*.

Two independent replicates (Rep 1 and Rep 2) for RNA samples of four different genotypes were barcoded and sequenced on a single HiSeq 2500 instrument. Yield (Mb) is the total number of sequenced bases. %PF (% passed filter) indicates the percentage of clusters (sequenced polynucleotides) that fulfill default Illumina quality criteria; Cluster (PF) indicates the number of such clusters. Q30 is the percentage of bases detected with a quality score greater or equal to 30. Mean quality is the average of quality scores for all called bases.

## Bioinformatic analyses

Illumina sequence reads were mapped to the *A. thaliana* Col-0 reference genome (TAIR10) using Bowtie v0.12.8 (*Langmead et al., 2009*). End-to-end best hits were reported allowing a maximum of one mismatch per read and up to 50 genomic mappings (for reads with multiple best hits). Reads corresponding to 45S rRNA gene arrays, chloroplast transcripts or mitochondrial transcripts were excluded at this stage. For each RNA size class and genomic position, the number of reads mapped with coinciding 5′ ends, oriented in the same direction was estimated. Read counts were then normalized by dividing by the total count of mapped reads for the sample and multiplying by the maximum total read count across all samples. For subsequent genomic analyses tRNAs, small nuclear RNAs (snRNAs) and small nucleolar RNAs (snoRNAs) were excluded from datasets based on their TAIR10 annotation coordinates. Diagrams of P4R2 RNA and siRNA spatial relationships were produced using FEATnotator v1.2.2 (*Podicheti and Mockaitis, 2015*). Sequence logos were generated with WebLogo v2.8.2 (*Crooks et al., 2004*).

RNA-seq reads of ~15–94 nt (after adapter trimming) were mapped to the *Arabidopsis thaliana* genome (version TAIR10) using Bowtie v0.12.8. The total number of reads used for analyses in the paper, which includes reads with 100% match to the genome or reads with a maximum of one mismatch, are provided.

## Accession numbers

RNA-seq data for the eight libraries generated in this study (*Blevins et al., 2015*) are available from the National Center for Biotechnology Information Sequence Read Archive (http://trace.ncbi.nlm.nih.gov/Traces/sra/?study=SRP059814)

## Protein immunoprecipitation

Protein extraction and affinity immunoprecipitation were performed as in (*Haag et al., 2012*). Frozen leaves (4.0 g) were ground in liquid nitrogen, homogenized in extraction buffer (300 mM NaCl, 20 mM Tris-HCl pH 7.5, 5 mM MgCl$_2$, 5 mM dithiothreitol [DTT], 1 mM phenylmethylsulfonyl fluoride [PMSF] and 1:200 plant protease inhibitor cocktail [Sigma]), filtered through two layers of Miracloth (Calbiochem, San Diego, CA), and centrifuged to pellet cellular debris. Supernatants were incubated with 25 µL anti-FLAG-M2 resin (Sigma, St. Louis, MO) for 3 hr at 4°C on a rotating mixer. Resin was recovered and washed three times with extraction buffer supplemented with 0.5% NP-40.

## DCL3 activity assay

Total RNA (~360 µg) was extracted from wild-type (Col-0) or *dcl2/3/4* triple mutant inflorescence tissue and size-fractionated (see 'RNA blot hybridization') to obtain substrates for DCL3 activity assays. Low-molecular-weight RNA was divided into 12-µg aliquots, which were then mixed with: protein extraction buffer (buffer control), FLAG-resin previously incubated with Col-0 protein extract (wild-type/resin control), or FLAG-resin previously incubated with cell-free extract from transgenic plants expressing the DCL3–FLAG protein (the DCL3 assay). These mixtures were then incubated at 30°C for 3 hr in dsRNA cleavage buffer (50 mM NaCl, 500 µM adenosine triphosphatase, 100 µM guanosine triphosphatase, 12.5 mM creatine phosphate, 0.2 U/µl creatine kinase, 600 µM MgCl$_2$, 0.4 U/µl RNaseOUT). RNA products were isolated from the reactions by phenol–chloroform extraction, and then precipitated by addition of 2 µl GlycoBlue (Life Technologies) and two volumes ethanol,

followed by centrifugation. The precipitated RNA was washed twice in 75% ethanol, resuspended in loading buffer and analyzed by RNA blot hybridization.

## Pol II and IV immunoprecipitation

Pols II and IV were immunoprecipitated from transgenic lines expressing NRPB2-FLAG or NRPD1-FLAG, respectively, as described previously (*Haag et al., 2012*). Briefly, 1 g of NRPB2-FLAG transgenic line leaf tissue or 4 g of NRPD1-FLAG transgenic line leaf tissue (for Pol IV) was ground to a powder in liquid nitrogen using a mortar and pestle and then resuspended in 3.5 ml extraction buffer (20 mM Tris-HCl, pH 7.6, 300 mM sodium sulfate, 5 mM magnesium sulfate, 5 mM DTT, 1 mM PMSF, 1% plant protease inhibitor cocktail [Sigma]) per gram of tissue. The resulting lysate was subjected to centrifugation at 16,000 × g for 15 min at 4°C and the supernatant was subjected to centrifugation again using the same conditions. 50 µl of anti-FLAG agarose resin was then added to the supernatant and incubated 3 hr at 4°C on a rotating mixer. The resin was collected by centrifugation at 200 × g for 2 min and the supernatant was discarded. Resin was washed twice with wash buffer (20 mM Tris-HCl, pH 7.6, 300 mM sodium sulfate, 5 mM magnesium sulfate, 5 mM DTT, 1 mM PMSF, 0.5% IGEPAL) and once with CB100 (25 mM HEPES-KOH pH 7.9, 20% glycerol, 100 mM KCL, 1 mM DTT, 1 mM PMSF). Between each wash, the resin was collected by centrifugation at 200 × g for 2 min and the supernatant was discarded. For Pol II, the washed NRPB2-FLAG resin was resuspended in a final volume of 1ml CB100 buffer. For Pol IV, the washed NRPD1-FLAG resin was brought to a final volume of 50 µl CB100.

## In vitro transcription assays

50 µl of washed NRPB2-FLAG or NRPD1-FLAG resin was mixed with 50 µl of 2 × transcription reaction mix (18 nM M13mp18 template [Bayou Biolabs, Metairie, LA], 2 mM each rNTP, 120 mM ammonium sulfate, 40 mM HEPES-KOH pH 7.9, 24 mM magnesium sulfate, 20 uM zinc sulfate, 20 mM DTT, 20% glycerol, 1.6 U/ul RNase inhibitor) and incubated at room temperature for 1 hr on a rotating mixer. Reactions were subjected to centrifugation through Performa spin columns (EdgeBiosystems, Gaitherburg, MD), the filtrate was adjusted to 0.3M sodium acetate, and nucleic acids were precipitated with isopropanol. The pellets were resuspended in 8 µl water and DNase-treated using Turbo DNA-*free* kit reagent (ThermoFisher Scientific, Grand Island, NY). The samples were then treated with Terminator exonuclease (Epicentre Biotechnologies, Madison, WI) to remove potentially degraded RNAs with 5' monophosphates, cleaned with RNA Clean and Concentrator kit (Zymo Research, Irvine, CA), and then treated with RNA 5' Polyphosphatase (Epicentre Biotechnologies) to convert 5' triphosphate groups to monophosphates. All enzyme treatments were performed according to the manufacturer's protocols. Adapter ligation and library construction was performed using a TruSeq small RNA kit (Illumina). Paired-end sequencing was conducted using an Illumina MiSeq instrument. Paired reads were mapped to the M13 sequence using default settings of BWA (*Li and Durbin, 2009*) for Illumina. Insert sizes were calculated using the CollectInsertSizeMetrics Picard tool. BWA and Picard tools were accessed via The Galaxy Project public server (UseGalaxy.org).

## Over-expression and purification of RDR2

RecRDR2 was overexpressed in insect cells using the BaculoDirect  baculovirus protein expression system (Thermo Fisher Scientific, Grand Island, NY). *RDR2* cDNA, amplified by PCR using primers 5'-CACCATGGTGTCAGAGACGACGACGAAC-3' and 5'-TCAAATGGATACAAGTCCACTTGTTTT-3', was cloned into pENTR and recombined into the baculovirus vector using LR clonase. The resulting baculovirus vector was transfected into Sf9 insect cells and recombinant virus was harvested according to the instructions of the manufacturer. RecRDR2 was purified from High Five cells infected with the recombinant virus and lysed in a buffer containing 50 mM HEPES-KOH (pH 7.5), 400 mM KCl, 5 mM DTT, 1 mM PMSF, 5 mM benzamidine-HCl and 10% glycerol. The lysate was subjected to centrifugation at 39,000 ×g for 30 min at 4°C. Anti-V5 agarose beads were used to selectively bind rec_RDR2 in the cleared lysate. The beads were collected by centrifugation at 170 × g for 5 min, washed 3 times with the same buffer, and eluted using V5 peptide in a buffer containing 50 mM HEPES-KOH (pH 7.5), 100 mM NaCl, 0.01% of NP-40 and 10% glycerol. The protein was concentrated and washed by buffer exchange using a Centricon column.

## RDR2 terminal transferase assay

RDR2 terminal transferase assays were performed according to (*Curaba and Chen, 2008*) with some modifications. RDR2 or RDR2-ASM (~250 ng) was added to activity buffer containing 20 mM HEPES-KOH (pH 7.5), 0.2 mCi/ml α-$^{32}$P-CTP, 5 mM MgCl$_2$, 5 mM DTT, 0.8 U/µl RNaseOUT and 1 µM of RNA template in a 50 µl reaction and incubated at room temperature (~24°C) for 60 min. Reactions were terminated by adding 10 mM ethylenediaminetetraacetic acid and then a clean-up step was performed using EdgeBio Performa (EdgeBiosystems, Gaithersburg, MD) spin columns. RNA products were ethanol precipitated and separated on a denaturing 15% polyacrylamide gel. The gel was transferred on Whatman 3MM filter paper and dried under vacuum at 80°C prior to phosphorimaging. The RNase treatment assay was conducted using a 20 µl RDR2 reaction with 0.05 U/µl of RNase ONE in 1 × reaction buffer. The reaction was incubated at 37°C for 30 min and terminated by 0.2% SDS .

## Acknowledgements

TB and CP designed the study. VM performed RDR2 overexpression and terminal transferase assays. TB, JW, and VM performed the exogenous DCL3 dicing assay. MM conducted in vitro transcription assays. TB performed all other experiments. RP conducted the bioinformatics analyses. TB and RP generated the graphics and figures and CSP wrote the manuscript. HT and DR provided guidance and advice for informatics analyses. The authors thank Jered Wendte for providing homozygous seed of the *ago4-7* mutant, and Ross Cocklin for performing the genetic cross to generate the *rdr2 dcl2/3/4* quadruple mutant.

## Additional information

### Funding

| Funder | Grant reference number | Author |
|---|---|---|
| National Institutes of Health | GM077590 | Craig S Pikaard |
| Howard Hughes Medical Institute | | Craig S Pikaard |
| Gordon and Betty Moore Foundation | GBMF3036 | Craig S Pikaard |

The funders had no role in study design, data collection and interpretation, or the decision to submit the work for publication.

### Author contributions

TB, RP, Conception and design, Acquisition of data, Analysis and interpretation of data, Drafting or revising the article; VM, DR, HT, Conception and design, Acquisition of data, Analysis and interpretation of data; JW, Acquisition of data, Analysis and interpretation of data; CSP, Conception and design, Analysis and interpretation of data, Drafting or revising the article

## Additional files

### Supplementary files

• Supplementary file 1. Table showing the coordinates and normalized read counts for 24 nt small interfering RNA (siRNAs) and P4R2 RNAs detected in wild-type (Col-0), *dcl2/3/4*, *pol IV* and *rdr2* within 100 bp windows. Data for two independent RNA sequencing (RNA-seq) replicates for each genotype are provided.

• Supplementary file 2. Pol IV/RDR2-dependent RNAs (P4R2 RNAs) overlap with Pol IV-dependent transcript loci identified by Li et al, 2015. P4R2 RNAs detected in the *dcl2 dcl3 dcl4* (*dcl2/3/4*) triple mutant were compared to a set of 22 Pol IV-dependent transcript loci verified by reverse transcription-polymerase chain reaction (RT-PCR) in the study of Li et al., 2015. The number of P4R2 RNAs ≥ 26 nt overlapping each Pol IV locus is shown for *dcl2/3/4* replicate 1 and 2 RNA sequencing (RNA-

seq) datasets. Unique reads are those that map to only one genomic location. Total reads include reads that can map to two or more genomic loci.

Major datasets

The following datasets were generated:

| Author(s) | Year | Dataset title | Dataset ID and/or URL | Database, license, and accessibility information |
|---|---|---|---|---|
| Blevins T, Podicheti R, Pikaard C | 2015 | Pol IV and RDR2-dependent pre-cursors of 24 nt siRNAs | http://trace.ncbi.nlm.nih.gov/Traces/sra/sra.cgi?study=SRP059814 | Publicly available at the NCBI Sequence Read Archive (Accession no: SRP059814). |
| Blevins T, Podicheti R, Pikaard CS | 2015 | Col-0 Rep 1 | http://www.ncbi.nlm.nih.gov/sra/SRX1070790 | Publicly available at NCBI Sequence Read Archive (Accession no: SRR2075815). |
| Blevins T, Podicheti R, Pikaard CS | 2015 | dcl2 dcl3 dcl4 Rep1 | http://www.ncbi.nlm.nih.gov/sra/SRX1070791 | Publicly available at NCBI Sequence Read Archive (Accession no: SRR2075816). |
| Blevins T, Podicheti R, Pikaard CS | 2015 | nrpd1-3 Rep 1 | http://www.ncbi.nlm.nih.gov/sra/SRX1282091 | Publicly available at NCBI Sequence Read Archive (Accession no: SRR2505369). |
| Blevins T, Podicheti R, Pikaard CS | 2015 | rdr2-1 Rep 1 | http://www.ncbi.nlm.nih.gov/sra/SRX1282116 | Publicly available at NCBI Sequence Read Archive (Accession no: SRR2505433). |
| Blevins T, Podicheti R, Pikaard CS | 2015 | Col-0 Rep 2 | http://www.ncbi.nlm.nih.gov/sra/SRX1070795 | Publicly available at NCBI Sequence Read Archive (Accession no: SRR2075819). |
| Blevins T, Podicheti R, Pikaard CS | 2015 | dcl2 dcl3 dcl4 Rep2 | http://www.ncbi.nlm.nih.gov/sra/SRX1070796 | Publicly available at NCBI Sequence Read Archive (Accession no: SRR2075821). |
| Blevins T, Podicheti R, Pikaard CS | 2015 | nrpd1-3 Rep 2 | http://www.ncbi.nlm.nih.gov/sra/SRX1282096 | Publicly available at NCBI Sequence Read Archive (Accession no: SRR2505408). |
| Blevins T, Podicheti R, Pikaard CS | 2015 | rdr2-1 Rep 2 | http://www.ncbi.nlm.nih.gov/sra/SRX1282131 | Publicly available at NCBI Sequence Read Archive (Accession no: SRR2505479). |

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
