## [Decision Letter]

Thank you for submitting your work entitled "Identification of Pol IV and RDR2-dependent precursors of 24 nt siRNAs guiding de novo DNA methylation in Arabidopsis" for peer review at *eLife*. Your submission has been evaluated by Jim Kadonaga (Senior editor) and three reviewers, one of whom is a member of our Board of Reviewing Editors. The reviewers feel the discovery that RNA Pol IV creates short transcripts is of major interest, as it substantially alters the existing model of RNA-directed DNA methylation. However, the reviewers expressed substantial concerns regarding the strength of the proof that the short RNA molecules described in the manuscript represent primary Pol IV transcripts.

The reviewers have discussed the reviews with one another and the Reviewing editor has drafted this decision to help you prepare a revised submission.

Essential revisions:

1) The evidence that short RNA molecules depend on Pol IV and RDR2 is provided for only one locus. This is insufficient evidence to claim that all such RNAs are made by Pol IV instead of Pol II or Pol V. Please provide additional evidence for Pol IV dependence, such as additional Northern blots or sequencing RNAs from POL IV dcl234 mutants. Depending on the strength of the additional evidence, the "P4R2" nomenclature may have to be reconsidered.

2) The data presented do not exclude the possibility that short RNAs are processed from longer primary Pol IV transcripts. Given the central conclusion of the paper, please strengthen the evidence that primary Pol IV transcripts are indeed short. This might be done through analysis of nascent transcripts, expanding the analysis that links short RNAs to Pol II consensus start sites that is briefly presented in the discussion, or through other approaches. Depending on the strength of the additional evidence, the authors may need to discuss the possibility that longer Pol IV transcripts are processed into the detected short molecules.

3) The data regarding Pol V dependence may be confusing by suggesting that all short RNAs depend on Pol V, whereas this is probably not the case. This could be resolved by probing the Northern for loci that do not depend on Pol V for 24-nt sRNA production (in conjunction with addressing point 1) or by a clear explanation in the text.

4) For the DCL3 in vitro assay, please include a time zero control to ensure that there were indeed larger RNAs present in the extracts from the dcl234 mutant plants that are processed by the Flag tagged DCL3 protein.

5) Please include information describing RDR2 expression and purification and remove the "data not shown" parts of the Results and Discussion.

6) The treatment of sRNA phasing is confusing. A new analysis is referenced in the Introduction, and lack of phasing could easily result from random Pol IV initiation or DCL3 cleavage. Please present the phasing analysis in the Results and consider discussing its significance in the Discussion section.

7) Please provide significance information (P values) of the motifs or the parameters used to identify the motifs presented in the manuscript.

8) Please define the "R" lane in Figure 6.

---

## [Author Response]

*1) The evidence that short RNA molecules depend on Pol IV and RDR2 is provided for only one locus. This is insufficient evidence to claim that all such RNAs are made by Pol IV instead of Pol II or Pol V. Please provide additional evidence for Pol IV dependence, such as additional Northern blots or sequencing RNAs from POL IV dcl234 mutants. Depending on the strength of the additional evidence, the "P4R2" nomenclature may have to be reconsidered.*

We thank the reviewers for this comment, as it has prompted us to expand our analyses and strengthen our story. To address this comment we have added four new figures or supplemental figures to the revised manuscript, namely Figure 5, Figure 5—figure supplement 1, [Supplementary-material SD1-data SD2-data], Figure 8. In the initial version of the manuscript, we focused solely on the comparison of RNA-seq datasets from wild-type (ecotype Col-0) plants and *dcl2 dcl3 dcl4* (abbreviated as *dcl2/3/4*) mutants, in which P4R2 RNAs accumulate to their highest levels. However, in parallel with these Col-0 and *dcl2/3/4* datasets, we had also performed RNA-seq analyses for *nrpd1* and *rdr2* mutants (with 2 biological replicates for each genotype). In the revised manuscript, we now make use of all of these datasets to show that the P4R2 RNAs that accumulate in *dcl2/3/4* mutants are also detected (albeit at lower levels) in wild-type Col-0 but are absent or depleted in *nrpd1* or *rdr2* mutants. A genome browser view making this point graphically for three representative loci is shown in new Figure 5. Figure 5—figure supplement 1 shows browser views for three additional loci. [Supplementary-material SD1-data] is a table showing more than 5500 genomic intervals (with a window size of 100 bp) in which 24 nt siRNAs are detected, and in which putative P4R2 RNAs (>25 nt, <95 nt) are also detected, in both *dcl2/3/4* and wild-type, but are depleted in *nrpd1* or *rdr2* mutants. In this table, data are provided for both independent biological replicates for each genotype, with high confidence P4R2 RNA – generating intervals being those in which the RNAs are depleted in both *nrpd1* and r*dr2* replicates. In [Supplementary-material SD2-data], we also show that 20 of the 22 Pol IV-dependent transcript loci confirmed by Li et al. (2015) to be capable of generating Pol IV transcripts are included among our P4R2 loci. These authors also provided genetic evidence for Pol IV and RDR2-dependence of the transcripts. Collectively, we think that these data show convincingly that P4R2 RNAs are, indeed, dependent on both Pol IV and RDR2, thus justifying the P4R2 nomenclature.

*2) The data presented do not exclude the possibility that short RNAs are processed from longer primary Pol IV transcripts. Given the central conclusion of the paper, please strengthen the evidence that primary Pol IV transcripts are indeed short. This might be done through analysis of nascent transcripts, expanding the analysis that links short RNAs to Pol II consensus start sites that is briefly presented in the discussion, or through other approaches. Depending on the strength of the additional evidence, the authors may need to discuss the possibility that longer Pol IV transcripts are processed into the detected short molecules.*

This request by the reviewers was not trivial. To address this concern as best we can, we have included data from a separate, ongoing study by student, and new author, Michelle Marasco. We show, as new Figure 8, that affinity-purified Pol IV makes relatively short transcripts when provided with a >7 kb long, single-stranded M13 virus DNA template. These transcripts are similar in size distribution to those of P4R2 RNAs observed in vivo. Moreover, the start site consensus we observed for P4R2 RNAs is also observed for these in vitro transcripts. In addition, the 3’ ends of the Pol IV transcripts also resemble the 3’ end sequences of P4R2 RNAs. These new data support the hypothesis that P4R2 RNAs are Pol IV transcripts.

*3) The data regarding Pol V dependence may be confusing by suggesting that all short RNAs depend on Pol V, whereas this is probably not the case. This could be resolved by probing the Northern for loci that do not depend on Pol V for 24-nt sRNA production (in conjunction with addressing point 1) or by a clear explanation in the text.*

We have clarified this point by adding a sentence to the Results section stating that *pol V* and *ago4* mutants most likely affect P4R2 and siRNA levels indirectly via their involvement in DNA methylation and histone modifications that have been shown to recruit Pol IV to target sites. We cite several relevant, recent papers.

*4) For the DCL3 in vitro assay, please include a time zero control to ensure that there were indeed larger RNAs present in the extracts from the dcl234 mutant plants that are processed by the Flag tagged DCL3 protein.*

Apparently, we did a poor job of explaining the controls for this figure, as these controls satisfy the reviewer (s)’s concern. We have re-written this section to make it clearer. The two controls (no added protein, or potentially FLAG-purified proteins from non-transgenic plants that don't express FLAG-tagged DCL3) were conducted side-by-side with the affinity-purified DCL3 sample, and they show that undigested precursor RNAs remain even after the incubation period allowed for DCL3 digestion. Thus the precursor RNAs must also have been present at time zero.

*5) Please include information describing RDR2 expression and purification and remove the "data not shown" parts of the Results and Discussion.*

We have added two new panels to what is now Figure 9, showing the purity of the recombinant RDR2. We have removed any reference to data not shown.

*6) The treatment of sRNA phasing is confusing. A new analysis is referenced in the Introduction, and lack of phasing could easily result from random Pol IV initiation or DCL3 cleavage. Please present the phasing analysis in the Results and consider discussing its significance in the Discussion section.*

Frankly, the phasing data are unnecessary, as it is obvious from our study that the dicing of precursors so short that they can only produce single siRNAs precludes the possibility of phasing. Thus we have deleted the figure panel in question and any discussion of phasing. This has helped keep the manuscript a manageable length while discussing the numerous new figures.

*7) Please provide significance information (P values) of the motifs or the parameters used to identify the motifs presented in the manuscript.*

We now provide bit scores for the sequence logos. We neglected to include them in the original version of the manuscript, which was an oversight.

*8) Please define the "R" lane in Figure 6.*

This is a lane showing end-labeled template RNA, as a size marker. This is now made explicit in the figure.